# Hybrid Photovoltaic/Thermoelectric Systems for Round-the-Clock Energy Harvesting

**DOI:** 10.3390/molecules27217590

**Published:** 2022-11-05

**Authors:** Yingyao Zhang, Peng Gao

**Affiliations:** 1College of Chemistry, Fuzhou University, Fuzhou 350108, China; 2State Key Laboratory of Structural Chemistry, Fujian Institute of Research on the Structure of Matter, Chinese Academy of Sciences, Fuzhou 350002, China; 3Xiamen Key Laboratory of Rare Earth Photoelectric Functional Materials, Xiamen Institute of Rare Earth Materials, Chinese Academy of Sciences, Xiamen 361021, China; 4Fujian Science & Technology Innovation Laboratory for Optoelectronic Information of China, Fuzhou 350002, China

**Keywords:** photovoltaic cells, thermoelectric generators, hybrid system, configuration, optimization

## Abstract

Due to their emission-free operation and high efficiency, photovoltaic cells (PVCs) have been one of the candidates for next-generation “green” power generators. However, PVCs require prolonged exposure to sunlight to work, resulting in elevated temperatures and worsened performances. To overcome this shortcoming, photovoltaic–thermal collector (PVT) systems are used to cool down PVCs, leaving the waste heat unrecovered. Fortunately, the development of thermoelectric generators (TEGs) provides a way to directly convert temperature gradients into electricity. The PVC–TEG hybrid system not only solves the problem of overheated solar cells but also improves the overall power output. In this review, we first discuss the basic principle of PVCs and TEGs, as well as the principle and basic configuration of the hybrid system. Then, the optimization of the hybrid system, including internal and external aspects, is elaborated. Furthermore, we compare the economic evaluation and power output of PVC and hybrid systems. Finally, a further outlook on the hybrid system is offered.

## 1. Introduction

In order to prevent further irreversible damage to the environment due to the uncontrolled use of fossil fuels, the search for clean, renewable, and environmental-friendly energy has become the focus of scientists worldwide. As mentioned by *Nikola Tesla*, “*Electric power is everywhere present in unlimited quantities and can drive the world’s machinery without the need of coal, oil, gas or any other of the common fuels*”, [1]. Therefore, fossil fuels should eventually be replaced by clean electric power. Hence, many technologies have emerged to convert various forms of energy in nature into electricity. For example, photovoltaic cells (PVCs) convert solar energy into electricity [2], Kaplan turbines and generators convert gravitational potential energy into electricity [3], wind turbines convert wind power to electricity [4], and thermoelectric generations (TEGs) directly convert thermal gradients into electricity [5].

The ubiquitous and inexhaustible nature of solar energy has attracted tremendous attention, and research on solar cells has grown dramatically. As shown in Figure 1a, the number of published papers about solar cells has increased yearly, especially after 2010. The working principle of PVCs can be summarized as the generation of carriers (including electrons and holes) when a cell is exposed to a radiation source, as indicated by the photoelectric effect [6]. With the continuous innovation and iteration of photoactive materials, PVCs have developed into various types [7,8,9], such as crystalline silicon solar cells [10,11,12,13], amorphous Si cells [14,15,16,17], and compound solar cells [18,19,20,21] (Details in Table 1). Among the various types of PVCs, crystalline silicon solar cells have proven to be the most commercially successful PVC technology due to their capability to provide high efficiency with affordable costs, and the great availability of silicon material on the earth [22].

Although the power conversion efficiency (PCE) of multi-tandem solar cells has surpassed 47%, [23] there is still a limit to the amount of incident photon energy that could be fully converted and utilized, according to Schockley–Queisser (SQ) limit theory [24]. The high energy photons beyond the bandgap of the semiconductor are relaxed to Joule heat energy, which heats up the PVC and reduces the photovoltaic (PV) efficiency while accelerating the cell degradation due to the collision of electrons in the depletion region of the solar cells [25,26]. Guo, X. et al. [27] assumed that although up to 85% of solar energy in the solar energy spectrum is absorbed by solar cells, more than 60% of that converts to heat instead of electrical energy. Moreover, solar cells usually suffer from temperature coefficients, and the performance downgrades with elevating temperatures. For example, approximately −0.3 to −0.39%/K for Si-based solar cells [28], −0.22%/K for perovskite [29], −0.21%/K for CdTe, and −0.36%/K for CIGS [30] have been recorded. Therefore, perovskite and CdTe solar cells with smaller temperature coefficients have the advantage of working under higher temperatures with minimized temperature-induced performance degradation. In brief, it is vital to cool down the photovoltaic cells via an appropriate method to maintain the performance of the equipment.

**Figure 1 molecules-27-07590-f001:**
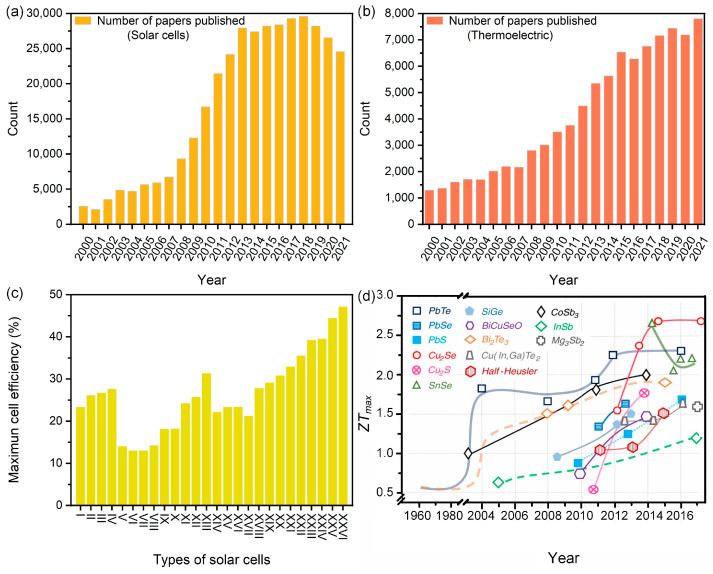
The number of papers published on (**a**) solar cells and (**b**) thermoelectric according to year. Source: Web of Science. Clarivate Analytic Data, https://www.webofscience.com/wos/, accessed on 15 July 2022. (**c**) The highest confirmed conversion efficiencies of research cells for a range of photovoltaic technologies. Label of solar cells as shown in Table 1. Source: National Renewable Energy Laboratory (NREL), accessed on 30 June 2022. (**d**) Timeline of the maximum *ZT* values for several representative families of TE materials. Reprinted with permission from [31] Copyright 2017, American Association for the Advancement of Science, AAAS.

It is worth mentioning that there are possibilities to combine two or three approaches to alleviate the disadvantages of a mono-system and increase overall energy utilization efficiency [32]. One typical example is the photovoltaic–thermal collector (PVT), containing solar panels with water or air channels passing from the backside. This method simply enables heat harvesting through the fluid and improves the solar energy utilization efficiency while controlling the temperature of PV modules. Moreover, the waste heat removed by the fluid can be used to heat boilers and for household heating, etc. In this vein, the considerable progress of thermoelectric (TE) materials and devices over the past decades provides new possibilities for solving the abovementioned problems (Figure 1b,d) (vide infra). TE generators vary based on the materials used for charge separation. For example, electronic TE generators use semiconductors [33] or conducting polymers [34,35], while ionic TE generators use liquid electrolytes [36], hydrogels [37,38,39], and ion gels [40]. As noted at the beginning, TEGs have the potential to directly convert a temperature difference into an electric potential difference through the thermoelectric (TE) effect, and vice versa, [5] which could be modeled as the Peltier effect, the Seebeck effect, and the Thomas Johann effect, depending on the actual working conditions [41]. For example, when a voltage is applied, Peltier modules can create a temperature difference between their two surfaces. Although Peltier modules can be used to solve the overheating of working PVCs, they have to consume extra electricity, which is not economical. Therefore, there has been considerable interest in cooling the PVC with Seebeck modules, which can harvest the waste heat accompanying the working PVC after sunlight exposure to generate electricity and act as additional energy-supplying devices.

The PVC–TEG hybrid system possesses several advantages over each mono-system. On the one hand, traditional PVCs rely on sunlight to produce electricity and cannot work at all at night. However, TE devices can work properly as long as there exists a temperature difference, meaning they are not as sensitive to the ambient environment as conventional PVCs. On the other hand, although TEGs have been applied as a power pack in many kinds of equipment, including medical devices, [42,43,44,45] wearable devices [46,47,48,49,50], and wireless sensor networks, [51,52] conventional portable TE devices may only generate power at the level of microwatts to milliwatts, which is not enough to support heavy-duty devices alone. Therefore, the hybrid system is not a simple superposition of the materials and costs, but provides a viable solution to significantly improve overall power output and realize genuine round-the-clock energy harvesting. In addition, the total electric power output is usually used as the standard figure of merit to evaluate hybrid systems. By comparing the overall power output between conventional solar cells and a hybrid system with the same area of solar cells, the advantages of hybrid systems can be visualized.

Hence, this review paper outlines the recent significant research on designing and optimizing PVC–TEG hybrid systems. Based on the working principle of both PVCs and TEGs, the optimization of both techniques is discussed from both internal and external perspectives. Furthermore, we compare the power output and costs between single PVCs and hybrid systems to analyze their relative technological and economic feasibility. Finally, we put forward the prospect of the future development of PVC–TEG hybrid systems.

## 2. Basic Principles of Solar Cells

Photovoltaic effects in solar cells take place in the presence of a p–n junction when the solar cells are exposed to sunlight. First, as shown in Figure 2a, photons with energy greater than the band gap (*E_g_*) of the absorbing material excite electrons in the valence band to the conduction band, leaving positively charged holes in the valence band, which are usually called photo-generated carriers. Then, the electrons in the conduction band and holes in the valence band diffuse to the space charge region of the p–n junction and are separated by the built-in electric field in this region. Therefore, the electrons are swept to the n-type side, and the holes are swept to the p-type side. Finally, positive and negative charges are accumulated on the upper and lower surfaces (poles) of the cells and generate photo-generated voltage. If the load is connected to both ends of the cells, a continuous current will pass through the load under continuous sunlight.

## 3. Basic Principles of Thermoelectricity

The temperature difference on both sides of the sample can result in the transport of charge carriers from the hot side toward the cold side, thus generating electrical voltage (vide infra). The TE performance is commonly parameterized by a dimensionless figure of merit, *ZT* [53,54,55,56]:(1)$$ZT=\frac{S^{2}\sigma T}{\kappa }$$
where *σ*, *S*, *κ*, and *T* are the electrical conductivity, Seebeck coefficient, total thermal conductivity, and absolute temperature, respectively. Hence, higher values of *S* and *σ* as well as simultaneously lower values of *κ* play a vital role in boosting *ZT*. Meanwhile, the power factor, $PF=S^{2}\sigma$, is also introduced to simply assess the TE efficiency of a material. Moreover, *σ* and *S* can be defined by:(2)σ=neμ
(3)$$S=\left(\frac{{8\pi }^{2}k}{3eh^{2}}\right)m\times T{\left(\frac{\pi }{3n}\right)}^{2/3}$$
where *n* is carrier concentration, *e* is the electron charge, *μ* is carrier mobility, *k* is the Boltzmann constant, *h* is the Planck constant, and *m** is the charge carrier mass. Obviously, *S* is inversely proportional to *n* which, in turn, is positively proportional to *σ*, and therefore, a trade-off between *S* and *σ* arises. In addition, *κ* consists of the electronic thermal conductivity (*κ_e_*) and the lattice thermal conductivity (*κ_L_*), as defined by:(4)$$\kappa =\kappa_{e}+\kappa_{L}$$
(5)$$\kappa_{e}=L\sigma T$$
(6)$$\kappa_{L}=\frac{1}{3}cvl$$
where *L* is the Lorenz number, *c* is the heat capacity, *v* is the velocity of sound, and *l* is the mean free path of phonons. As mentioned above, electrical conductivity, the Seebeck coefficient, and thermal conductivity are interrelated. Hence, it is impossible to increase the *ZT* value of a material monotonically during the optimization of all the parameters.

Figure 2b shows a typical schematic diagram of a TEG, in most cases composed of p-type and n-type materials, leading to the formation of the π-type TE element. When a temperature gradient is applied to both sides of the device, the charge carriers (electrons for n-type semiconductors and holes for p-type semiconductors) will transfer from the hot side toward the cold side and thus generate current. Furthermore, to accumulate the small electric potential generated by a single TE element (microwatt to milliwatt), a TE module consisting of a number of TE elements electrically in series and thermally in parallel is fabricated (Top of Figure 2b).

## 4. The Principle and Basic Configuration of the PVC–TEG Hybrid System

Depending on whether the sunlight diverges or not, hybrid systems can be divided into two versions, including integrated hybrid systems and spectrum-splitting hybrid systems [57,58,59]. In the former, the system is exposed to full-wavelength sunlight, while the latter splits the wavelengths of sunlight spectra into two segments for PVC and TEG, respectively.

Figure 3a shows the schematic configuration of the integrated hybrid system, which mainly consists of a solar concentrator, PVC, TEG, and heat sink. After the sunlight is collected by a solar collector, the PVC converts this solar energy into electricity and Joule heat. Then, the TEG directly fitted on the back side of the PVC converts the waste heat into electric power. By contrast, to make optimal use of the long-wavelength part of the spectrum, a solar-selective module was introduced to form a spectrum-splitting hybrid system (Figure 3b). In this system, the long-wavelength part of the spectrum will not only cause higher temperatures but will also be fully utilized due to the spectral-selective solar module that divides sunlight between the PVC and TEG. In addition, although models that connect the PVC and TEG in series, such as Ambimax [60], Everlast [61], Heliomote [62], Prometheus [63], and PUMA [64], have been proposed and designed, and have been applied in hybrid electric vehicles [65], these tandem systems are independent from hybrid systems, and so they will not be discussed in this review.

Previous work has confirmed that the hybrid system results in higher productivity and lower energy consumption [66,67,68]. For instance, one study set the concentrated photovoltaic thermal (CPVT) system as the control group, and compared it with a concentrated photovoltaic thermal—thermoelectric (CPVT–TE) hybrid solar system, as shown in Figure 3c [67]. In both systems, we exchanged water as a heat sink to maintain the temperature for the cold side of the TE module. Additionally, the exchanged water collected the waste heat for self-heating, which can be used in daily life. In addition, for a more visual comparison of the two systems, both were put into service in Tunisia and the results were compared. The eventual outcomes are presented in Figure 3d–f. Not only was the electrical efficiency enhanced with the hybrid system, but the energy consumption and the CO_2_ emission also decreased, which revealed that hybrid systems are superior in terms of energy savings and power supply. In summary, according to the experiment, it is feasible and practical to replace the PVT system with a hybrid one.

**Figure 3 molecules-27-07590-f003:**
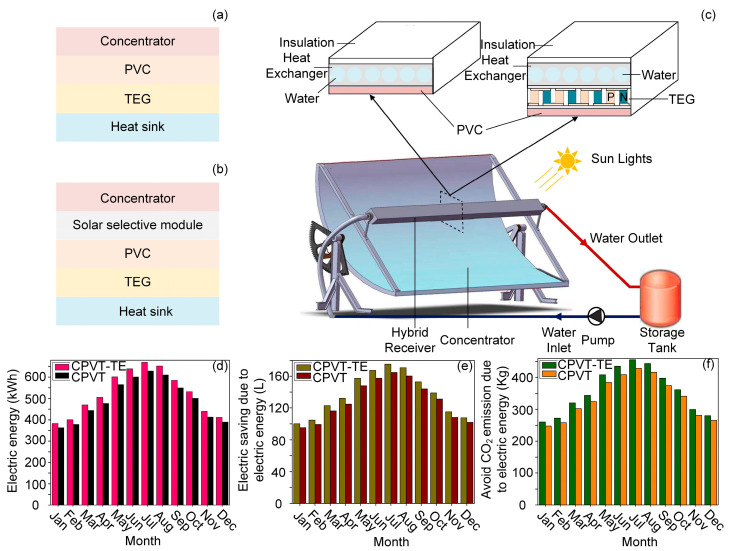
Schematic configuration of (**a**) integrated hybrid system and (**b**) spectrum splitting hybrid system. (**c**) The actual design of the hybrid solar system and the cross-sectional schematics of the receiver. Reprinted with permission from [67]. Copyright 2019, Elsevier. The inset shows the exploded view of the PV module. Monthly distribution of (**d**) the electric energy, (**e**) the energy saving of gasoline, and (**f**) the avoided CO_2_ emission due to electric energy production of the CPVT and CPVT–TE system. Reprinted with permission from [67]. Copyright 2019, Elsevier.

## 5. The Optimization of the Hybrid System

In order to obtain higher power output, optimization of the hybrid system is essential. According to the two versions of the hybrid system (vide supra), they can be optimized with respect to both internal and external aspects. The former focuses on the system, which can be divided into three key factors: materials, geometry, and structure. The latter is to adjust the incident light and transmit the light with the appropriate wavelength to the corresponding modules.

### 5.1. Internal Optimization

#### 5.1.1. Materials

##### Shielding Materials

Unlike in controlled experimental conditions, various environmental factors should be considered in actual operating conditions, for example, wind, rain, dust, and corrosion, which may impact the system’s performance. Covering the surface of a hybrid system with a layer of glass may seem to be a simple and effective protective method. Although the glass can protect the system from deterioration due to the ambient environment and decrease the thermal loss by air convection, the efficiency of glassless hybrid system is always higher than that of one with glass [69,70,71]. In order to overcome the trade-off between efficiency and protection, a glass cover with high transmissivity has to be used to overcome the efficiency difference between these two systems [25,69]. Additionally, organic polymer materials could also be used to encapsulate PVCs. As shown in Figure 4a, a solar cell was placed between ethylene vinyl acetate (EVA) films and polyethylene terephthalate (PET), which not only ensured the output efficiency but also isolated the device from the surrounding environment. In addition to EVA and PET, poly(methylmethacrylate) (PMMA) is another choice for such an enclosure [72] (Figure 4b).

##### Energy Conversion Materials

Next, various types of PVCs, for example, polycrystalline silicon and dye-sensitized solar cells (DSSC), can lead to different performance of the hybrid system [73]. The hybrid system with polycrystalline silicon as the absorbing material in PV cells generated higher output and performed better than DSSC. Interestingly, at higher temperatures, although the maximum output of the DSSC-based configurations is still lower than that of the polycrystalline-silicon-solar-cell-based devices, the performance improvement for DSSC-based systems was significant, demonstrating that DSSC technology could become particularly attractive under conditions of elevated temperature. Moreover, in order to further optimize the overall efficiency of the hybrid system, the criteria of optimization for all the parameters have to be confirmed, e.g., the effects of temperature on the performance of the DSSC-TEG hybrid system [74]. In fact, except for Si-based solar cells [32,75,76] and DSSCs, other kinds of solar cells have also been widely studied in hybrid systems, including GaAs solar cells [77], CIGS solar cells [78], and perovskites solar cells [79,80,81]. In addition, ZnO nanowire-based light-trapping technology can be applied to enhance the efficiency of PVCs [78].

Besides the dominant PVCs in the hybrid system, the accompanying TEGs can significantly improve overall efficiency in some cases [22]. Hence, the optimization of TEGs plays a vital role in optimizing hybrid systems. However, unlike the various absorbing materials available for PVCs, bismuth telluride (Bi_2_Te_3_) is the only commercial room-temperature thermoelectric material. A perspective on alternatives to conventional TE materials is proposed in the outlook section (Section 6).

##### Thermal Management Materials

Further, a decreased thermal resistance between PVCs and TEGs is critical to keep the PVCs at a lower temperature while maintaining the efficiency of the TEG. Therefore, high-thermal-conductivity materials, such as ceramic and copper, are used to decrease thermal resistance. For example, inlaying solar cells on a copper plate with the proper thickness could enhance the heat transfer to the TEGs underneath, thus resulting in a high-performance hybrid system [72] (Figure 4b). Furthermore, thin nano-CuO film, which possesses superior thermal conductivity, could be used to replace the copper plate for further optimization [82,83]. In addition, with copper as the container, phase-change materials (PCMs) were introduced to suppress the temperature increase after exposing PVCs to sunlight, and to maintain the hybrid system at the optimal operating temperature [77]. The PCM is a material that changes its shape with temperature and can provide latent heat. In the process of changing phase, the material will release or absorb heat, and the objects in contact with it will maintain a constant temperature. Moreover, the PCM can minimize the thermal resistance between PVCs and TEGs, leading to efficient mutual heat transfer.

Finally, the heat sink, conventionally a fluid such as water, mercury, or liquid potassium, determines the temperature of the cold side of the TEG by dissipating heat to maintain the temperature gradient, which is essential to the efficiency of TEG [84]. Likewise, nano-fluid was also confirmed to be able to improve the overall efficiency of the hybrid system [69]. Therefore, an effective thermal transfer between the TEG and the heat sink should be considered. Besides the traditional heat dissipation materials (copper and ceramic), a high-density polyethylene (HDPE) matrix containing aluminum (Al) powder and internal microstructure has also been used as the functionally graded material (FGM) interlayer to increase heat transfer [76].

**Figure 4 molecules-27-07590-f004:**
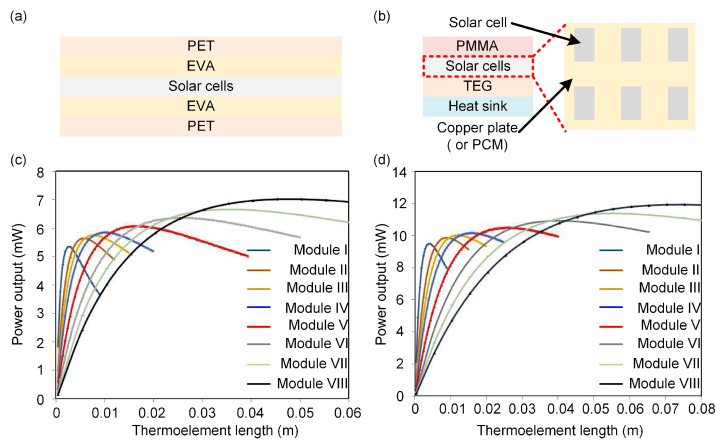
Schematic illustration of (**a**) covered PVC and (**b**) novel mosaic structure of PVC. Power output as a function of thermo-element length for system operation at (**c**) ambient and (**d**) vacuum atmosphere. Reprinted with permission from [85]. Copyright 2016, Elsevier.

#### 5.1.2. Geometry

Despite the importance of thermal management materials between PVCs and TEGs (vide supra), most commercial TEGs lack an ideal thermal resistance for hybrid systems [86]. Therefore, the geometry of the TEGs should be optimized to tune the cost, thermal resistance, and internal resistance [87,88,89,90,91]. Resistance-matching plays a crucial role in achieving maximum power output in a hybrid system, and this is usually optimized by the geometry of TEGs. Although it is challenging to achieve synergistic effects when simply combining two different systems, significant effort has been dedicated to adjusting TEGs to realize gains in total energy harvesting [32,89,90]. For example, Figure 4c,d shows the power output of TE modules with different numbers of thermo-elements, different cross-sections, and different thermo-element lengths under ambient atmospheric or vacuum conditions (details in Table 2) [85]. In either case, with a continuous increase in thermo-element length, the power output rises monotonically to reach the maximum before it starts to decrease. Shorter thermo-elements can provide lower operating temperatures for the PVCs, due to the lower thermal resistance of the shorter TE module [73]. It is worth mentioning that the shorter thermo-elements also lead to lower economic costs.

Moreover, with the same length and number of thermo-elements, a TE module with a smaller cross-section area possesses higher power output than those with a larger cross-section area. In addition, when the length of the thermo-element is low, the generated electrical power is inversely proportional to the number of thermo-elements, while it becomes a diametrically opposite situation with a longer thermo-element length. In summary, depending on the specific parameters of the hybrid system, the number and length of thermo-elements do not simply amount to ‘the more (longer), the better’ [69]. It is also worth mentioning that higher power output can be obtained if the system operates under a vacuum, originating from the decrease in the convective heat losses.

#### 5.1.3. Structure

In order to further increase the power output of the hybrid system, novel construction is needed. A new parabolic trough structure was designed to replace the traditional parallel structure, containing a triangular or semicircular channel with an outer surface covered with PVCs and the TE modules directly mounted on its back (Figure 5a) [71,84]. The parabolic trough reflects sunlight on the PVC and, compared with the traditional method of adding a concentrator on the top of the PVC, in this case, PVCs receive sunlight from more directions rather than just vertical direction. In addition, the triangular structure not only increases the light-receiving area but also is more suitable for parabolic troughs, which is beneficial to enhance the overall efficiency of the hybrid system.

Moreover, various thermosyphons (e.g., glass, copper, stainless steel, and nickel) have been introduced to replace the traditional heat sink to maintain the temperature of the cold side of the TEG and carry the remaining thermal energy to a bottom cycle (Figure 5b) [84]. The thermosyphon is a high-efficiency passive two-phase heat transfer element consisting of an evaporator section and a condenser section. Firstly, the sunlight is reflected to the PVCs by the parabolic trough to warm up the PVCs. Then, the heat is transferred to the TEG and the heat sink (thermosiphon), where the liquid working medium absorbs the latent heat and transforms into a gaseous state to reach the condenser section. Then, the gaseous working medium releases the latent heat through the wall to the external cold source. After that, the gaseous working medium is transferred into the liquid state and transported back to the evaporator section under gravity to absorb the latent heat and circulate continuously. The thermosiphon has the advantages of a simple structure, easy manufacture, high heat transfer efficiency, stable working condition, wide working temperature range, and low maintenance cost.

Notably, increasing the output efficiency of the TEG cannot rely solely on lowering the temperature of the cold side. Increasing the temperature of the hot side should also be taken into account. A structure of absorption–conduction–insulation was designed to absorb more solar energy and convert it into heat for the TEG. The conductive layer can be attached to the side and back of the PVC, after which it is covered by the absorptive layer, before finally attaching the insulation layer (Figure 5c) [75].

On the one hand, the absorbing layer collects solar radiation energy; on the other hand, the incident light can be divided into infrared and visible light. Therefore, the PVC absorbs only the visible wavelength in the sunlight and reflects the infrared light to the absorbing layer. Then, the heat flux flows from the absorbing layer to the conducting layer, which leads to a higher temperature gradient on both sides of the TEG, thereby optimizing the output efficiency of the TE module. In addition, the insulating layer is used to effectively prevent heat loss caused by heat exchange.

### 5.2. Optical Management

One important strategy for optimizing the hybrid system is the management of optical paths of the solar spectrum. In general, the most efficient spectral response region of the solar cell is the photons with energy close to the solar cell band gap. Photons with energy higher or lower than the band gap can only be partly utilized: (i) any given photovoltaic material is transparent to the photons with energy below its band gap; (ii) the solar cell is heated up by excess energy from the photons above its band gap. In order to make complete utilization of solar energy in the broad-solar-spectrum wavelength range, the notion of spectrum-splitting hybrid systems was developed. Previous work has shown that the broad solar spectrum could be partitioned into different segments for PVCs and TEGs, which each possess their respective optimal absorption wavelength ranges [92]. Therefore, spectrum splitting seems an effective method to segment the spectrum and realize the full potential of hybrid systems [93,94].

Based on the various spectrum-splitting techniques proposed for solar cells [95,96,97,98], researchers have developed similar techniques for the hybrid system. Five main spectrum-splitting systems are discussed as follows:(i)Prisms [99]: refractively disperse sunlight and concentrate different portions of the spectrum onto PVC (Figure 6a);(ii)Filters [100,101]: only transmit radiation of a suitable range to the solar cells and suppress unwanted solar radiation (Figure 6b);(iii)Luminescent solar concentrators [102,103,104]: concentrated sunlight enters the second component. Then, in the first segment (S1), some photons are absorbed by the fluorophores and re-emitted isotropically to the PVC, while others may propagate further according to the device and be absorbed by the following segments (S2, S3, etc.), until the unwanted part is discharged through S4 (Figure 6c);(iv)Dichroic mirrors [96]: concentrate the sunlight by hollow pyramid concentrators and then separate the spectrum by dichroic mirrors to solar cells with different energy gaps (Figure 6d);(v)Holographic solar concentrators and spectrum splitters [105,106,107]: diffract the far-infrared light, while the near-infrared light is spectrally separated to optimized solar cells, as shown in Figure 6e.

In brief, the techniques mentioned above split the sunlight gathered by the concentrator in different ways into specific wavelengths to be collected by PVCs with an appropriate bandgap. Such external optimization not only makes full use of sunlight but also addresses the root causes of the overheating problem. Interestingly, a novel structure of PVC was designed, which possesses high absorption at the wavelength of 300–1100 nm and high transmission at the wavelength of 1100–2500 nm [108]. The novel structure ensures the high performance of PVCs and prevents the heating up of solar cells by the long-wavelength part of the spectrum, which instead strengthens the output of TEGs.

For an ideal hybrid system, the following parts should be included: concentrator, spectral splitter, PVC, TEG, and heat sink. Among them, the concentrator will be used to collect sunlight for conversion into electricity by solar cells. Therefore, parabolic concentrators and PVC structures with larger areas (Figure 5a,b) are more dominant. Then, the spectrum splitter will transmit the collected sunlight to solar cells with different band gaps to ensure that sunlight is fully utilized and that sunlight of unsuitable wavelengths will affect PVC as little as possible. After that, the TEG will be directly fit on the back side of solar cells, which converts the Joule heat generated by the PVC into electricity with the help of heat sinks. Furthermore, the heat generated by sunlight can also be collected by the TEG to generate more electricity, as shown in Figure 5c. In addition, the hybrid systems should be encapsulated with high-transmittance materials to protect the device without affecting its performance.

## 6. Economic Evaluation

In order to further analyze the feasibility of a hybrid system, the cost-effectiveness of a single crystalline PVC and a hybrid system will be discussed (Table 3 and Table 4) [76]. According to Table 3, compared with traditional PVCs, the actual PVC efficiency in the hybrid system is increased by 1.2%. Interestingly, perovskite solar cells can obtain 3.05% [109] and 4.9% [80] efficiency gain with hybrid systems, which demonstrates that perhaps perovskite solar cells are more suitable for such tandem systems. Based on the total electric power output (Table 3), it is possible to evaluate the hybrid system. The overall electric output of the hybrid system is increased by 24.51% compared to conventional single-Si cells, which demonstrates the superiority and feasibility of the hybrid system. Although the total power output increased after the PVC was coupled with the TEG, due to the high price of TEGs, the overall cost of a hybrid system will increase by about 90% (Table 4), leading to a convenient but uneconomical hybrid system [110,111]. Nevertheless, with the continued development of TEGs, although the high cost may currently limit the hybrid system, the emergence of more and more low-cost TE materials will gradually solve this problem; we will put forward our perspective on these development in the following outlook.

## 7. Outlook

In order to mark current trends and the future potential of development, Figure 7a demonstrates histograms of relevant papers published over the years, with data prior to 2005 not counted due to the small number. The increasing number of yearly publications indicates that the hybrid system is gaining importance. We believe this trend will continue because of its unique characteristics of making the best use of solar energy. Figure 7b indicates the internal working mechanism of the hybrid system. Firstly, solar cells generate electricity after being exposed to sunlight. Most of the losses out of the Shockley–Quisser limit are in the form of heat, which can then be utilized by TEGs to generate electricity.

Interestingly, easily generating accessible electricity for Internet of Things (IoT) devices is also a hot topic, such as powering medical sensors and power supplies for unmanned areas. The Internet of Things (IoT), known as the “Internet connecting everything”, is an extended network that enables the interconnection of people, machines, and things at any time and from any place. The hybrid system is the best candidate for IoT energy supply due to its stability and long maintenance intervals. Furthermore, heat sinks are essential since TEGs require a temperature difference between the two ends. The thermal management material (e.g., water) in the heat sink can also be used for heating, enabling a complete utilization of energy in the hybrid system.

The performance of a TE material is defined by its figure of merit *ZT*, and thus the TE modules composed of a material with higher *ZT* have the potential to improve the overall efficiency of the hybrid system. As the first compound semiconductor thermoelectric material, the Bi_2_Te_3_-based alloy is the best TE material near room temperature and is the most widely studied TE material. With its outstanding TE performance and simplicity of preparation, it has occupied the dominant position in the commercial application of thermoelectric materials for more than half a century. Fortunately, over the past decade, many materials with superior *ZT* values have been discovered, for example, SnSe [33,112,113,114,115,116,117,118], Cu-based materials [119,120], and Mg-based materials [121,122]. Furthermore, organic semiconductors (OSC) have recently attracted attention with their unique characteristics enabling lightweight, flexible, and intrinsically low thermal conductivity [123,124,125,126]. Although the maximum *ZT* value of organic thermoelectric materials is just over 0.2 [124,127,128,129], which is far below that of inorganic thermoelectric materials, it is still an option for manufacturing low-weight, low-toxicity, and flexible TEGs. Organic thermoelectric generators (OTEGs) are not limited by the shape of PVCs due to their flexibility. Moreover, OTEGs are more suitable for mass production, as they are generally prepared from precursor solutions. This leads to the preparation process being greatly simplified, for example, by ink painting and spraying.

In addition to hybrid systems using PVC–TEG tandems, the advent of thermoradiative cells (TRCs), which belonged to energy-emissive harvesters and were proposed in 2014 [130], can also be used to replace TEGs. The working principle of TRCs is similar to that of PVCs, and they can be considered anti-solar cells. Solar cells use the sun as the hot end and itself as the cold end. By contrast, TRCs use themselves as the hot side and point toward a cold object [131,132]. Band diagrams and the path of carriers for PVCs and TRCs under different operating conditions will be discussed as follows:
(i)Under the conditions of thermal equilibrium and without illumination (Figure 8a), the absorbed photons of PVCs were equal to the emitted photons, where the quantity of total electrons (*n*) and holes (*p*), intrinsic electrons (*n_0_*), intrinsic holes (*p_0_*), and the total intrinsic carrier density (*n_i_*) conform to $np=n_{0}p_{0}=n_{i}^{2}$, and the position of the Fermi level (*E_F_*) remains constant.(ii)Under the illumination conditions (Figure 8b), the absorption of photons by solar cells is greater than the emission from cells and thus generates photovoltage (vide supra). The generation of photo carriers leads to $np>n_{0}p_{0}=n_{i}^{2}$ and splits *E_F_*.(iii)By contrast, when the p−n junction functions as a TRC (Figure 8c), emission from the solar cells is greater than absorption, resulting in $np<n_{0}p_{0}=n_{i}^{2}$ and the *E_F_* splits in the opposite direction.

It is worth noting that both TRCs and PVCs generate usable power, and the main difference between them is that the current flows and the voltage signals are reversed. With the characteristic that TRCs can generate electricity without sunlight, a hybrid system that combines solar cells and TRCs has the potential to generate power around the clock. Additionally, when using TRC devices, deep space is the best cold end, but the thermal heat must be able to be emitted to outer space [131,133].

According to the Shockley–Queisser framework in PVCs, the theoretical conversion limit of TRCs has been calculated to be 33.2%, with a TRC with an energy gap of 0.25 eV at a temperature of 500 K in surroundings at 300 K [132,133]. Therefore, suitable materials for TRC application should have extremely small bandgaps of less than 0.3–0.4 eV and a low rate of non-radiative recombination [134]. In order to enhance the power output and efficiency of TRCs, intermediate-band thermoradiative cells (IB-TRCs) have been introduced, which contain an intermediate-band material between two semiconductors. Besides the transition from the conduction band to the valance band, IB-TRCs also allow for inter-band transitions from the conduction band to the intermediate band and from the intermediate band to the valance band, which ultimately leads to a larger electric current due to the low-energy electrons contributing to the current generation [135,136]. However, a prerequisite for IB-TRCs is that the intermediate electronic band must have a dispersion in energy that is big enough to consider the group of states as a band, and not as a set of impurity levels, because the impurity levels will induce a strong Shockley–Read–Hall (SRH) recombination. This is complicated to realize in a narrow-band gap of 0.3–0.4 eV without high SRH rates [134,137,138,139]. In addition, similar to the optical management strategy in solar cells (vide infra), selectively emitting only low-frequency photons can significantly improve the efficiency of TRC [140].

## 8. Conclusions

This review identifies the structure of hybrid systems composed of PVCs and TEGs. Then, in order to optimize the power output of hybrid systems, we summarized the measures for both internal and external optimizations. Internal optimization improves the system, for example, its materials, geometry, and structure. By contrast, external optimization is to adjust the solar spectrum, splitting the spectrum into two segments for PVCs and TEGs, which enhances the overall efficiency and decreases the temperature of solar cells. Finally, we proposed that OTEGs, with their unique characteristics of flexibility and simple preparation processes, can be economically applied to various unconventional shapes of PVC. In addition, we expect that TRCs can be used to replace TEGs as a candidate for next-generation hybrid systems.

## Figures and Tables

**Figure 2 molecules-27-07590-f002:**
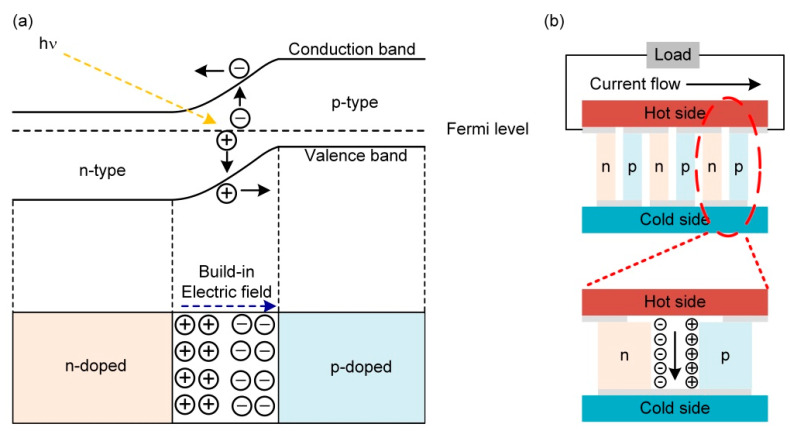
Schematic diagrams of (**a**) photovoltaic effect and (**b**) a TE module (**top**) and a TE element (**bottom**).

**Figure 5 molecules-27-07590-f005:**
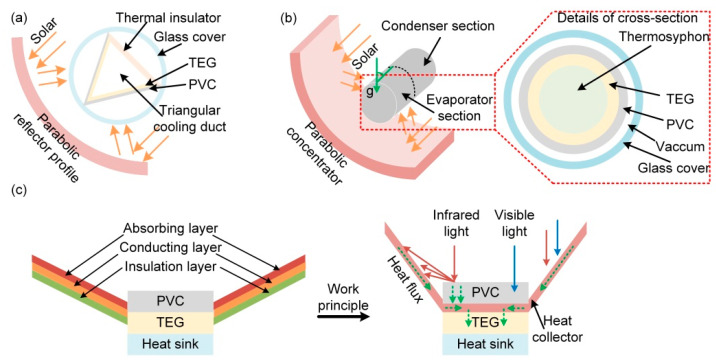
Schematic illustration of (**a**) triangular, (**b**) thermosyphon-technology-based, and (**c**) absorption–conduction–insulation hybrid systems.

**Figure 6 molecules-27-07590-f006:**
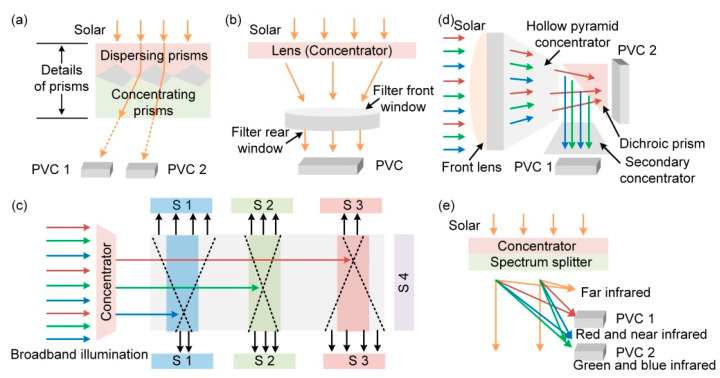
Schematic of (**a**) refractive spectrum-splitting module of prism arrays and PVC arrays, and details of prism arrays, (**b**) filter, (**c**) luminescent spectrum splitter, (**d**) fabricated dichroic mirrors, and (**e**) holographic concentrator and spectrum splitter.

**Figure 7 molecules-27-07590-f007:**
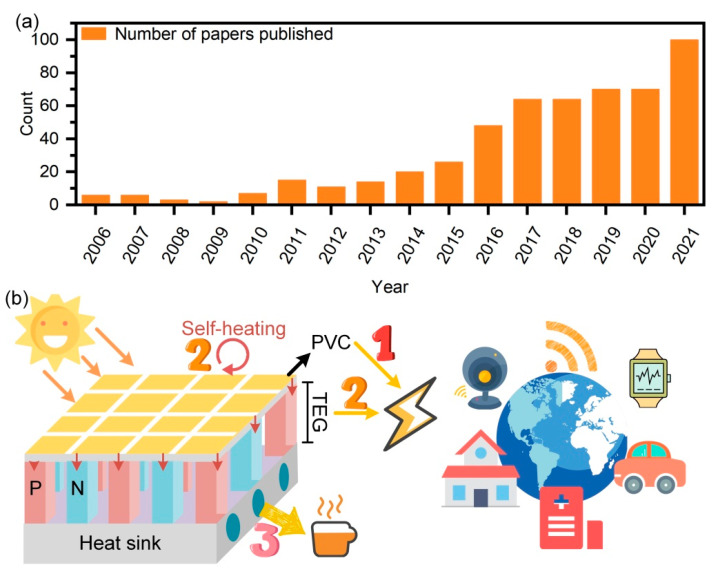
(**a**) The number of papers published on PVC–TEG hybrid (tandem) systems organized by year. Source: Web of Science. Clarivate Analytic Data, https://www.webofscience.com/wos/, accessed on 17 October 2022. (**b**) Schematic diagram of energy flow and application for the hybrid system.

**Figure 8 molecules-27-07590-f008:**
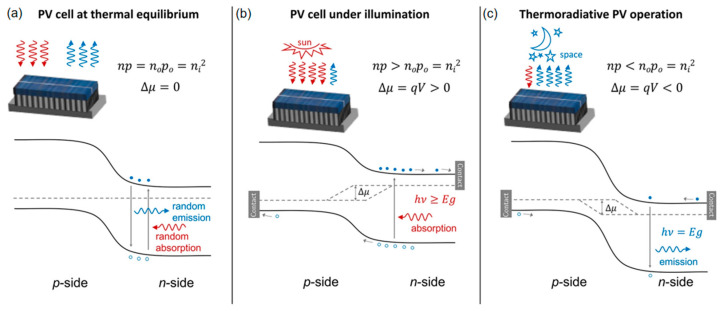
The band diagrams and the path of carriers for (**a**) PVCs at thermal equilibrium, (**b**) PVCs under illumination, and (**c**) TRC operation. Reprinted with permission from [131]. Copyright 2020, American chemical society.

**Table 1 molecules-27-07590-t001:** The highest confirmed conversion efficiencies and categories of different solar cells. Source: National Renewable Energy Laboratory (NREL), https://www.nrel.gov/pv/cell-efficiency.html, accessed on 30 June 2022.

Solar Cells		Label	Maximum Cell Efficiency (%)
Crystalline Si cells	Multicrystalline	I	23.3
	Single crystal (non-concentrator)	II	26.1
	Single heterostructures (HIT)	III	26.7
	Single crystal (concentrator)	IV	27.6
Amorphous Si cell	Amorphous Si:H (stabilized)	V	14
Compound cells	Inorganic cells (CZTSSe)	VI	13
	Dye-sensitized cells	VII	13
	Organic tandem cells	VIII	14.2
	Quantum dot cells (various types)	IX	18.1
	Organic cells	X	18.2
	Perovskite/CuInGaSe_2_ (CIGS) tandem (monolithic)	XI	24.2
	Perovskite cells	XII	25.7
	Perovskite/Si tandem (monolithic)	XIII	31.3
	CdTe	XIV	22.1
	CIGS (concentrator)	XV	23.3
	CIGS	XVI	23.4
	Thin-film crystal	XVII	21.2
	Single crystal GaAs	XVIII	27.8
	Concentrator GaAs	XIX	29.1
	Thin-film crystal GaAs	XX	30.8
	Two-junction (non-concentrator)	XXI	32.9
	Two-junction (concentrator)	XXII	35.5
	Four-junction or more (non-concentrator)	XXIII	39.2
	Three-junction (non-concentrator)	XXIV	39.5
	Three-junction (concentrator)	XXV	44.4
	Four-junction or more (concentrator)	XXVI	47.1

**Table 2 molecules-27-07590-t002:** Geometric parameters of the experimental TE modules.

Module Type	Number of Thermo-Elements	Cross-Section Area (mm^2^)
I	62	0.64
II	62	1.44
III	62	1.96
IV	62	2.56
V	100	2.56
VI	150	2.56
VII	200	2.56
VIII	250	2.56

**Table 3 molecules-27-07590-t003:** Performance analysis of the traditional PVC hybrid system.

System Types	Actual PV Cell Efficiency (%)	Total Electric Power Output (W/m^2^)
Single crystalline Si cells	9.1	91
Hybrid system	10.3	113.3

**Table 4 molecules-27-07590-t004:** Cost analysis of the hybrid system.

Item	Materials	Hybrid System Cost ($/m^2^)
Substrate	Fire-retardant plywood	6.78
FGM layer	Aerated polyvinyl chloride	67.24
	Al powder	22.38
	Extruding tubes	2.69
TE layer	Bulk Bi_2_Te_3_ module	1678.25
PV layer	Single crystalline Si cells	193.64
Superstrate	EVA	9.47
	PV glass	28.51
Perimeter	Al clip	5.7
	Piping materials	13.23
	Pumping system	65.84
Total cost	None	2095.88

## Data Availability

Not applicable.

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
