# Peer review of "Hybrid Photovoltaic/Thermoelectric Systems for Round-the-Clock Energy Harvesting"

_molecules, 2022, doi:10.3390/molecules27217590_

Round 1

Reviewer 1 Report

In this review, the authors describe the principles of photovoltaic and thermoelectric systems separately, and detail the advantages of hybrid systems. I think this review is very suitable for readers to have a preliminary understanding of the PVC-TEG hybrid system, and can provide more technical support for the development of this industry. I suggest accepting this manuscript after a major revision.

1.     In line 93-103, the authors claim that the PVC/TEG hybrid system possesses a series of advantages that are far better than the mono system. However, to my knowledge, it is difficult to say 1+1>2 when combining two different systems. It usually requires balances, optimizations, and compromise. So how do researchers overcome such problems? What are the parameters that are set as the standard figure of merit for the hybrid system?

2.     In line 182, “precious works”?

3.     In line 182-193, the authors introduce their hybrid system for convincing the advantages of the hybrid system. It would be more convincing if the results of the comparison were directly based on significant data differences in the text.

4.     In Table 1 and section 5, the authors introduce Si based PVC and DSSC. How about other types of solar cells, such as perovskites? Are there any hybrid systems designed based on these?

5.     In section 5, the authors introduce several routes to design and optimize the hybrid system. I have a question. What is the best design of the hybrid system to date? What is the most distinctive feature, and what makes it the best design?

Author Response

Reply to the reviewers

First of all, I would like to express our sincere gratitude to the reviewers for the comments. These comments are all valuable and helpful for revising and improving our manuscript, as well as the crucial guiding significance to our research. We have read the comments carefully and made corrections, which we hope meet with approval. Revision portions are marked in red in the revised version. The summary of corrections and the responses to the Reviewer’s comments are listed below.

Summary of the revision:

  • Section 1: We added some information about the temperature coefficient of solar cells, the state-of-the-art of thermoelectric generators, and the standard figure of merit for the hybrid system.
  • Section 4: We corrected the spelling mistakes and added a more precise and concise statement for Figure 3 d-f.
  • Section 5: We added some information on combining two different systems, and the conception of an ideal hybrid system.
  • Section 7: We added some information about thermoradiative cells, and significant data differences between solar cells and hybrid system.
  • Responses to reviewers (original comments by reviewers are in blue)
  •  
  • Reviewer#1:
  • In this review, the authors describe the principles of photovoltaic and thermoelectric systems separately and detail the advantages of hybrid systems. I think this review is very suitable for readers to have a preliminary understanding of the PVC-TEG hybrid system, and can provide more technical support for the development of this industry. I suggest accepting this manuscript after a major revision.
  • Thank you for your appreciation of our work and for giving constructive comments to improve the quality of this manuscript. We carefully revised the manuscript according to the valuable suggestion.
  • Comment: In line 93-103, the authors claim that the PVC/TEG hybrid system possesses a series of advantages that are far better than the mono system. However, to my knowledge, it is difficult to say 1+1>2 when combining two different systems. It usually requires balances, optimizations, and compromise. So how do researchers overcome such problems?
  • Reply: Thank you for the constructive comments. We have revised this sentence for a more explicit expression in Section 5.1.2, lines 290 to 294.
  • “The resistance matching plays a crucial role in achieving maximum power output in a hybrid system, which is usually optimized by the geometry of TEGs. Although it is challenging to achieve synergistic effects when simply adding up two different systems, significant efforts have been dedicated to adjusting the TEGs to realize the gains in total energy harvesting.[1–3] Figure 4c,d show the power output of TE modules with different numbers of thermo-elements, cross-section, and thermo-element lengths under ambient or vacuum atmosphere. (Details in Table 2)[4]”
  •  
  • Figure 4. Power output as a function of thermo-element length for system operation at (c) ambient and (d) vacuum atmosphere. Reprinted with permission from[4]. Copyright 2016, Elsevier.
  •  
  • Table 2. Geometric parameters of the experimental TE modules.

Module type

Number of thermo-elements

Cross-section area (mm2)

I

62

0.64

II

62

1.44

III

62

1.96

IV

62

2.56

V

100

2.56

VI

150

2.56

VII

200

2.56

VIII

250

2.56

  •  
  • Comment: What are the parameters that are set as the standard figure of merit for the hybrid system?
  • Reply: It is a constructive question. We have added the relative discussion in Section 1, lines 112-116.
  • “In addition, the total electric power output is usually used as the standard figure of merit to evaluate the hybrid system. By comparing the overall power output between conventional solar cells and a hybrid system with the same area of solar cells, the advantages of hybrid systems can be visualized.”
  •  
  • Comment:In line 182, “precious works”?
  • Reply: We are sorry for this mistake that should not appear, and we have corrected it in Section 4, line 190.
  • “Previous works have confirmed that the hybrid system possesses higher productivity and lower energy consumption”.
  • Comment:In line 182-193, the authors introduce their hybrid system for convincing the advantages of the hybrid system. It would be more convincing if the results of the comparison were directly based on significant data differences in the text.
  • Reply: We appreciate the suggestion. We have mentioned this in Section 4 and Section 6, lines 197-203 and 415-418.
  • But it may not be clear enough, so we added a more precise and concise statement. i) “In addition, for a more visual comparison of the two systems, both were put into service and conducted in Tunisia. The eventual outcomes are presented in Figure 3 d-f. Not only was the electrical efficiency enhanced with a hybrid system, but the energy consumption and the CO2 emission also decreased, which revealed that hybrid systems are superior in terms of energy savings and power supply. In summary, according to the experiment, it is feasible and practical to replace the PVT system with a hybrid one.”
  • ii) “Based on the total electric power output,(Table 3) it is possible to evaluate the hybrid system. The overall electric output of hybrid system is increased by 24.51% compared to conventional single Si cells, which demonstrates the superiority and feasibility of the hybrid system. ”
  •  
  • Table 3. Performance analysis of the traditional PVC hybrid system.

System types

Actual PV cell efficiency (%)

Total electric power output (W/m2)

Single crystalline Si cells

9.1

91

Hybrid system

10.3

113.3

  •  
  • Comment: In Table 1 and section 5, the authors introduce Si based PVC and DSSC. How about other types of solar cells, such as perovskites? Are there any hybrid systems designed based on these?
  • Reply: We thank the Reviewer for the insightful comments. Different types of solar cells are used in the hybrid system and are briefly described in Section 5.1.1, lines 246-250.
  • “Actually, Except for the Si-based solar cells [1,7,8] and DSSCs, GaAs solar cells[9], CIGS solar cells[10], and perovskites solar cells[6,11,12] have also been widely studied in a hybrid system. In addition, ZnO nanowires-based light-trapping technology can be applied to enhance the efficiency of PVCs.[10].”
  • Comment: In section 5, the authors introduce several routes to design and optimize the hybrid system. I have a question. What is the best design of the hybrid system to date? What is the most distinctive feature, and what makes it the best design?
  • Reply: It is a good question. We put our opinions about the best design of the hybrid system in Section 5, lines 396-407.
  • “For an ideal hybrid system, the following parts should be included, concentrator, spectral splitter, PVC, TEG, and heat sink. Among them, the concentrator will be used to collect sunlight for conversion into electricity by solar cells. Therefore, parabolic concentrators and PVC structures with larger areas (Figure 5a, b) are more dominant. Then, the spectrum splitter will transmit the collected sunlight to solar cells with different band gaps to facilitate that sunlight is fully utilized and ensure that the sunlight of unsuitable wavelengths will affect PVC as little as possible. After that, TEG will be directly fit on the back side of solar cells, which converts the Joule heat generated by PVC into electricity with the help of heat sinks. Furthermore, the heat generated by sunlight can also be collected by TEG to generate more electricity, as shown in Figure 5c. In addition, the hybrid systems should be encapsulated with high transmittance materials to protect the device without affecting its performance.”

Reviewer 2 Report

In this review article, the authors discussed the development of hybrid photovoltaic/thermoelectric systems, in which the thermoelectric module could have three roles: cooling down the solar cell; making use of the photothermal effect of the solar cell; generating electricity at night time. They summarized the systems by starting from the basic principle of PVCs and TEGs and discussed the principle and basic configuration of hybrid system. The optimization direction from both internal and external aspects, are analyzed. Therefore, I think it is a very comprehensive work and crucial review article for the field. Plus, there is not much work like this being published recently. Therefore, I suggest publication after some revisions as suggested below:

1.       Different types of solar cells have different temperature coefficient. Could the author provide a discussion about the temperature coefficient and emphasize the different impact on the solar cells.

2.       The record efficiencies of PVs have been mentioned. It is encouraging to also mention the state-of-the-arts of different thermoelectric generators, e.g. 10.1126/science.aaz5045; 10.1126/science.abn8997; 10.1039/D1TA10508F; 10.1021/acs.jpclett.2c00845; 10.1002/asia.202200850; 10.1002/aenm.202200858

3.       At the end of the review, it is necessary to collect more information about the thermoradiative photovoltaics. Currently this technology is promising to realize night time working solar cell.

Author Response

Reply to the reviewers

First of all, I would like to express our sincere gratitude to the reviewers for the comments. These comments are all valuable and helpful for revising and improving our manuscript, as well as the crucial guiding significance to our research. We have read the comments carefully and made corrections, which we hope meet with approval. Revision portions are marked in red in the revised version. The summary of corrections and the responses to the Reviewer’s comments are listed below.

Summary of the revision:

  • Section 1: We added some information about the temperature coefficient of solar cells, the state-of-the-art of thermoelectric generators, and the standard figure of merit for the hybrid system.
  • Section 4: We corrected the spelling mistakes and added a more precise and concise statement for Figure 3 d-f.
  • Section 5: We added some information on combining two different systems, and the conception of an ideal hybrid system.
  • Section 7: We added some information about thermoradiative cells, and significant data differences between solar cells and hybrid system.
  •  
  • Reviewer#2:
  • In this review article, the authors discussed the development of hybrid photovoltaic/thermoelectric systems, in which the thermoelectric module could have three roles: cooling down the solar cell; making use of the photothermal effect of the solar cell; generating electricity at night time. They summarized the systems by starting from the basic principle of PVCs and TEGs and discussed the principle and basic configuration of hybrid system. The optimization direction from both internal and external aspects, are analyzed. Therefore, I think it is a very comprehensive work and crucial review article for the field. Plus, there is not much work like this being published recently. Therefore, I suggest publication after some revisions as suggested below:
  • Thank you for your positive comments on this work. We have read the comments carefully and made corrections.
  • Comment: Different types of solar cells have different temperature coefficients. Could the author provide a discussion about the temperature coefficient and emphasize the different impact on the solar cells.
  • Reply: The suggestion is valuable. We have added the relative discussion in Section 1, lines 61-67.
  • “Moreover, solar cells usually suffer from temperature coefficients and the performance downgrades with elevating temperature. For example, approximately -0.3 to -0.39%/K for Si-based solar cells[13], -0.22%/K for perovskite[14], -0.21%/K for CdTe, and -0.36%/K for CIGS[15] have been recorded. Therefore, perovskite and CdTe solar cells with smaller temperature coefficients have the advantage of working under higher temperatures with minimized temperature-induced performance degradation”.
  • Comment: The record efficiencies of PVs have been mentioned. It is encouraging to also mention the state-of-the-art of different thermoelectric generators, e.g., 10.1126/science.aaz5045; 10.1126/science.abn8997; 10.1039/D1TA10508F; 10.1021/acs.jpclett.2c00845; 10.1002/asia.202200850; 10.1002/aenm.202200858
  • Reply: Thank you for the very detailed advice. We have added this paragraph in Section 1, lines 88-91.
  • “TE generators varied based on the materials used for the charge separation, for example, electronic TE generators using semiconductors[16] or conducting polymers[17,18], while ionic TE generators using liquid electrolytes[19], hydrogels[20–22], and ion gels[23].”
  • Comment: At the end of the review, it is necessary to collect more information about the thermoradiative photovoltaics. Currently this technology is promising to realize night time working solar cell.
  • Reply: We thank the Reviewer for the constructive opinion. We have added the information about thermoradiative cells in Section 7, lines 469-470 and 495-513.
  • i) “which belong to energy-emissive harvesters and was proposed in 2014”, ii) “According to the Shockley-Queisser framework in PVCs, the theoretical conversion limit of TRC has been calculated to be 33.2%, when a TRC with an energy gap of 0.25eV at a temperature of 500K in surrounding at 300K.[24,25]. Therefore, suitable materials for TRC application should have extremely small bandgaps of less than 0.3-0.4 eV and a low rate of non-radiative recombination.[26] In order to enhance the power output and efficiency of TRC, the intermediate-band thermoradiative cells (IB-TRC) are introduced, which contain an intermediate-band material between two semiconductors. Besides the transition from the conduction band to the valance band, IB-TRCs allow the inter-band transitions from the conduction band to the intermediate band and from the intermediate band to the valance band, which lead to a larger electric current due to the low-energy electrons contributing to the current generation.[27,28] However, a prerequisite for IB-TRC is that the electronic intermediate electronic band must have a dispersion in energy that is big enough to consider the group of states as a band and not a set of impurity levels because the impurity levels will induce a strong Shockley-Read-Hall (SRH) recombination. It is complicated to realize in a narrow band gap of 0.3-0.4 eV without high SRH rates.[26,29–31] In addition, similar to the optical management strategy in solar cells (vide infra), selectively emitting only low-frequency photons can significantly improve the efficiency of TRC.[32]”

Round 2

Reviewer 1 Report

I think this version is good enough to be published.